# Protocol for a Randomized Crossover Trial to Evaluate the Effect of Soft Brace and Rigid Orthosis on Performance and Readiness to Return to Sport Six Months Post-ACL-Reconstruction

**DOI:** 10.3390/healthcare11040513

**Published:** 2023-02-09

**Authors:** Sonja Jahnke, Caren Cruysen, Robert Prill, Fabian Kittmann, Nicola Pflug, Justin Amadeus Albert, Tibor de Camargo, Bert Arnrich, Aleksandra Królikowska, Anna Kołcz, Paweł Reichert, Łukasz Oleksy, Sven Michel, Sebastian Kopf, Michael Wagner, Sven Scheffler, Roland Becker

**Affiliations:** 1Center of Orthopaedics and Traumatology, University Hospital Brandenburg a.d.H., Brandenburg Medical School Theodor Fontane, 14770 Brandenburg a.d.H., Germany; 2Faculty of Health Sciences Brandenburg, Brandenburg Medical School Theodor Fontane, 14770 Brandenburg a.d.H., Germany; 3Rehathleticum GmbH Functional Diagnostic Physiotherapy Center Berlin, 10587 Berlin, Germany; 4Digital Health-Connected Healthcare, Hasso Plattner Institute, University of Potsdam, 14482 Potsdam, Germany; 5Ergonomics and Biomedical Monitoring Laboratory, Department of Physiotherapy, Faculty of Health Sciences, Wroclaw Medical University, 50-367 Wroclaw, Poland; 6Department of Trauma Surgery, Faculty of Medicine, Wroclaw Medical University, 50-367 Wroclaw, Poland; 7Department of Physiotherapy, Faculty of Health Sciences, Jagiellonian University Medical College, 31-008 Krakow, Poland; 8Department of Therapy Sciences, Brandenburg University of Technology Cottbus-Senftenberg, 03046 Cottbus, Germany; 9Sporthopaedicum Berlin, Center for Orthopedics, Sports Medicine and Orthopedic Surgery, 10627 Berlin, Germany

**Keywords:** anterior cruciate ligament reconstruction (ACLR), return to sport (RTS), Back in Action (BIA), orthosis, brace, surface electromyography (sEMG), dynamic knee valgus (DKV), hip abductor muscle strength

## Abstract

A randomized crossover trial was designed to investigate the influence of muscle activation and strength on functional stability/control of the knee joint, to determine whether bilateral imbalances still occur six months after successful anterior cruciate ligament reconstruction (ACLR), and to analyze whether the use of orthotic devices changes the activity onset of these muscles. Furthermore, conclusions on the feedforward and feedback mechanisms are highlighted. Therefore, twenty-eight patients will take part in a modified Back in Action (BIA) test battery at an average of six months after a primary unilateral ACLR, which used an autologous ipsilateral semitendinosus tendon graft. This includes double-leg and single-leg stability tests, double-leg and single-leg countermovement jumps, double-leg and single-leg drop jumps, a speedy jump test, and a quick feet test. During the tests, gluteus medius and semitendinosus muscle activity are analyzed using surface electromyography (sEMG). Motion analysis is conducted using Microsoft Azure DK and 3D force plates. The tests are performed while wearing knee rigid orthosis, soft brace, and with no aid, in random order. Additionally, the range of hip and knee motion and hip abductor muscle strength under isometric conditions are measured. Furthermore, patient-rated outcomes will be assessed.

## 1. Introduction

The incidence of anterior cruciate ligament (ACL) tears has been reported as 68.6 per 100,000 cases per year [1], with a re-rupture rate of 1–11% after primary ACL reconstruction (ACLR) [2]. Long-term functional deficits are common in re-ruptures, e.g., chronic knee instability, meniscus tears, and cartilage damage [3,4,5]. In the long term, 50% of patients develop osteoarthritis after an ACL tear [6]. In addition, a failed ACLR is related to a rapid increase in cartilage degeneration [7]. Furthermore, ACL tears are associated with muscle atrophy, leading to a loss of functionality and stability of the knee [8]. Moreover, there is evidence that only about 50% of post-ACLR patients regain their preinjury athletic level [9]. In the worst case, this leads to the end of an athlete’s career [2]. Finally, it should also be considered that athletes with ACL tears have long rehabilitation periods and have to bear significant psychological burdens [10], which can highly influence their quality of life [11].

Multimodal programs are already established and focus on strength, flexibility, proprioception, and balance training. In addition, feedback training, understanding of biomechanics, and correct landing patterns are becoming increasingly important in the rehabilitation process [3,12]. Nevertheless, return to sport (RTS) programs show limited prospective value in terms of reinjuries [13]. Furthermore, there are no gold standard criteria for a safe RTS after ACLR [14]. Meanwhile, studies have shown that in addition to assessing quantitative criteria during the rehabilitation process, it is also essential to examine qualitative criteria such as dynamic knee valgus (DKV) [15], which increases during jump landing and thus increases the risk of an ACL injury [16]. In summary, this is the rationale for further investigation to identify the barriers and factors to successful rehabilitation and to bring athletes back to their preinjury sporting performance.

Less than ten years ago, the Back in Action (BIA) test battery, a standardized tool for deciding whether a patient is ready for RTS after an ACLR was developed and published [17]. The test battery consists of a double-leg stability test, a single-leg stability test, a double-leg countermovement jump (CMJ), single-leg CMJ, a plyometric jump test consisting of three plyometric jumps, a speedy jump test, and a quick feet test [17]. When analyzing the results, gender, patient age, and leg dominance are considered, providing an overview of the patient’s complete functional knee performance [17]. Studies have demonstrated that the BIA test battery is superior in showing side-to-side differences compared to other RTS test batteries [18], e.g., the standard hop and strength tests [19]. Therefore, introducing BIA in the decision-making process for a proper RTS is necessary to assess functional recovery and dynamic knee stability [19].

Knee orthotic devices are often recommended and used after ACLR. However, it is neither proven if there is a mechanical advantage of using them nor which orthotic device has the most effective influence on body balance and coordination and, thus, the function of the knee joint [20]. In a recently published pilot study that the present research team conducted on young participants without a history of knee problems, it was shown that rigid orthoses and soft braces have neither a positive nor a negative influence on the BIA test battery results [21]. It was concluded that in healthy subjects, knee coordination and knee function are not influenced by using orthotic devices. The risk of injury or accident in sports does not seem to increase further, as no influence on knee coordination has been demonstrated. It remains unclear whether there is a preventive effect or minimization for the risk of injury and a resulting advantage for patients with ACLR. A systematic review showed that the influence of prophylactic knee bracing in American football does not seem to reduce the incidence of knee injuries [22]. As it is currently not conclusively proven whether there are any advantages or disadvantages using orthotic devices, this study will assess the aspect of athletic performance during the BIA test battery.

Soft or rigid orthotic devices may support feedback strategies through skin contact and thus increase the subjective feeling of safety, resulting in a willingness to perform. In addition, braces and orthoses provide patients with security (even if only psychologically) but are not mechanically proven. The immediate clinical relevance is that orthotic devices either positively or negatively influence the tested performance on RTS or not.

For sporting performance and benefits to be taken from orthotic devices, the feedforward and feedback strategies model is becoming increasingly important. Feedforward and feedback control are generally explained as mechanisms to restore homeostasis in an organism [23]. Therefore, feedback can be described as a response to a sensory detection (e.g., unstable joint position during landing maneuvers), and using previous experience (e.g., jumping exercises) to make an appropriate modification (e.g., adjustment of knee position). On the other hand, feedforward control describes the preparatory action that occurs before an event. Experiences are initially brought into pre-activation. When repeatedly performing a jump, the athlete can better estimate how much force must be used to optimize the performance. Restoration of functional stability in the knee joint occurs through unconscious dynamic control in preparation for and in response to the movement [23]. Therefore feedforward, as an anticipatory strategy, and feedback, as an action occurring in response to the sensory detection, are helpful strategies to achieve the movement goal and optimize training in rehabilitation [23]. Feedforward is mainly based on experience with an expected task. A good reproduction of real effects occurs when no specific preparation for a complex task is carried out, because the more complex the task, the less feedforward mechanisms can take effect [23,24].

Surface electromyography (sEMG), is an objective method to analyze muscular activation and coordination patterns during the execution of movements [25]. Cross et al. examined the EMG activity of the semitendinosus and gracilis muscles six months after they were harvested for ACLR and found no significant decrease in EMG activity [26]. However, after harvesting the semitendinosus graft for ACLR, regeneration of the muscle takes up to one year [27]. Therefore, it would be interesting to evaluate a possible difference in muscle activation between the limbs under sporting conditions six months after surgery.

Furthermore, the role of gluteus muscles on DKV is not fully clarified [28]. Due to the controversial discussion a possible correlation between muscle activation patterns and DKV will be investigated.

Motion tracking systems are a common instrument to track postural control [29]. The Microsoft Azure Kinect DK can be used to analyze the knee axis during some of the BIA test battery components. The methods provide the therapist and the patient with objective feedback on the development of performance in rehabilitation [30,31].

Furthermore, device-based strength measurements are considered the gold standard for assessing muscle function [32]. Therefore, they are useful tools for evaluating strength status during rehabilitation, especially when used in interventional designs [33]. A recent study showed that weak hip muscles in hip abduction, extension, and external rotation impact DKV in collegiate female athletes in single-leg tasks [34].

To the best of our knowledge, there have been no studies conducted to investigate the following aspects using the specifically designed BIA test battery, yet:The influence of rigid orthoses and soft braces on the performance of the knee jointThe activation of the gluteus medius and semitendinosus muscles and the associated influence on the cumulative DKV during landing maneuversConclusions on feedforward and feedback mechanisms during the test battery

### Objectives and Hypothesis

In this study, we will perform the functional and biomechanical assessment of patients six months (+/−1 month) after unilateral primary ACLR using an autologous hamstring tendon graft with a test battery, using either a knee rigid orthosis, knee soft brace (medi^®^ company, Medi Bayreuth, Germany) or no aid.

Our aim is to investigate the influence of muscle activation on functional stability/control of the knee joint to determine whether bilateral imbalances still occur six months after successful ACLR and to analyze whether the use of a brace or orthosis changes the activity onset of these muscles.

Furthermore, conclusions on the feedforward and feedback mechanisms are also highlighted.

In the event of significant lateral limb differences in muscle activity and DKV emergence persisting six months after ACLR, the results give reason to explore strategies to optimize the rehabilitation process, implement adapted exercises to reduce lateral limb differences, and improve the outcome of knee functionality to reduce the risk of a reinjury.

In addition, conclusions will be drawn about the influence and possible effectiveness of using orthotic devices and the associated benefits for patients. Based on the results, coaches, athletes, and medical staff need to have (no) concerns that individual use of supports/orthoses will affect athletic performance and RTS test results, which may help athletes who like to continue wearing aids for psychological reasons to decide on their usefulness. Based on the research questions, it can be hypothesized:

**Hypothesis** **1:**
*Soft braces or rigid orthoses improve functional performance during the modified BIA test battery.*


**Hypothesis** **2:**
*Muscle activity of the gluteus medius and semitendinosus muscles is impaired in the ACL-reconstructed leg during the modified BIA test battery.*


**Hypothesis** **3:**
*Impairment of gluteus medius and semitendinosus muscle function is related to lower performance in the operated limb in the modified BIA test battery.*


**Hypothesis** **4:**
*An impairment of gluteus medius strength and semitendinosus muscle activation correlates with DKV and pelvic drop emergence during the modified BIA test battery.*


## 2. Methods

Ethical approval is currently under consideration in the county medical chamber as an extension to a previously approved ethical application. The processing number is 22-TEMP897788-BO.

The assessment will be conducted in the Rehathleticum, a functional diagnostic physiotherapy center in Berlin (Germany).

A signed informed consent form and the confirmation of personal data protection will be obtained. Participants can withdraw from the study at any time without giving a reason. All data is anonymized by assigning a patient number so that a later assignment of personal data is only possible via the Sporthopaedicum in Berlin and the University Hospital of Brandenburg, Center for Orthopaedics and Traumatology. All measures are taken to protect the privacy of the subjects and the confidentiality of personal data.

### 2.1. Study Design

This study is a randomized, prospective, single-blind clinical trial with a three-group cross-over design. Due to the design of the study, complete blinding is not possible. Statistical analysts are blinded on the groups to be compared. The randomization procedure and allocation are done through a person who was not further involved. SJ and CC are responsible for the test setup and finally carry out the tests.

Patients are only informed about the performance but not about the target character of the tests. After receiving written informed consent, three patients are randomly assigned to either group A, B, or C (Table 1) and therefore run the test battery with rigid orthosis, soft brace and without aids in a cross-over order.

The bioelectrical activity of the gluteus medius and semitendinosus muscles are recorded using sEMG according to Surface ElectroMyoGraphy for the Non-Invasive Assessment of Muscles (SENIAM) guidelines [25]. Surface electrodes are placed on the involved and uninvolved limbs.

### 2.2. Participants

Participants will be recruited at the Sporthopaedicum, Center for Orthopaedics, Sports Medicine and Orthopaedic Surgery in Berlin (Germany) and the University Hospital of Brandenburg, Center for Orthopaedics and Traumatology (Germany). Suitable patients are selected via a file review and invited for a surveillance test. The test battery is performed six months (+/−1 month) postoperatively to observe the greatest possible difference while reducing the risk of reinjury to a minimum.

The studied cohort of 28 patients consists of female and male patients. Only a large impact on performance is internally valid for estimating whether braces or orthoses should be worn as they are used to reduce adverse events. For this reason, a potential cut off for wearing braces or orthoses was set at a large estimated effect. A large effect for the investigated three degrees of freedom, with an acceptable alpha error of 0.05 and a 1-beta error of 0.9, results in a required total sample size of 27 when using an ANOVA: fixed effects, special, main effects, and interactions model. To catch up one possible drop out, which is not expected, a sample size of 28 is included.

#### 2.2.1. Inclusion Criteria

six months (+/−1 month) after post-traumatic primary unilateral arthroscopically assisted ACLR, with the use of autologous ipsilateral semitendinosus graft;no additional procedures including meniscus sutures during the arthroscopically assisted ACLR;no osteoarthritis surgery during the arthroscopically assisted ACLR other than shaving;no history of injury or disease in the ACL-reconstructed limb prior to the ACL injury within the last 6 months;no history of injury or disease in the adjacent joints, contralateral limb, or spine within the last 6 months;age 18–40 years;Body Mass Index (BMI) smaller or equal to 24.9 kg/m^2^;moderate activity with regard to the rehabilitation plan, physiotherapy, and training therapy;frequent recreational sporting activity prior to the ACL injury.

#### 2.2.2. Exclusion Criteria

secondary ACLR;ACLR with the use of a method other than autologous ipsilateral semitendinosus tendon graft;diagnosed additional systematic diseases;current impaired performance due to previous injuries apart from the ACL injury;current permanent knee pain or swelling.

### 2.3. Procedure

The examination is performed in a single session, including three runs of the modified BIA test battery.

The patients are asked to abstain from unaccustomed strenuous exercise for at least 24 h before the testing and avoid eating a heavy breakfast in the morning before the test and no rich meals within two hours of the test. They are advised to wear comfortable sports outfits and indoor sports shoes. Only the double-leg and single-leg stability tests are carried out barefoot.

SEMG electrodes are applied to all participants on the ACL-reconstructed and unaffected legs on gluteus medius and semitendinosus muscles to measure muscle activation throughout the whole test battery.

After block randomization to determine the group, participants complete the test battery as presented in Table 2. For fatigue reasons, the unaffected limb is only tested in the first run. The best performance from each trial is selected for analysis.

## 3. The Extended Back in Action Battery Protocol

### 3.1. Balance Test

Patients perform a double-leg stability test and single-leg stability tests on the Zebris Balance Force Plate, starting with the non-injured leg. While maintaining stability for 10 s on one leg with arms crossed in front of the chest, the contralateral leg needs to be flexed and is not allowed to touch the ground. If balance is lost, the test has to be repeated [36]. The center of pressure is calculated and the sEMG records the muscle activity for a later calculation of time to peak, onset, offset and EMG force-relation.

### 3.2. Jumps

#### 3.2.1. General Criteria

One practice trial with submaximal power (80% of the maximal power) is given, followed by three subsequent test trialsHands are placed on the hips during the jumpsStarting position in the center of the plate, with feet parallel and shoulder-width apartKnee and hip joints are extended during the flight phaseSoft landing in the center of the 3D force plate with knee over toe position, stable leg axisSingle-leg landing: the contralateral leg is not allowed to touch the ground, otherwise the test is stopped immediately

All tests are explained in detail and demonstrated by the tester. The subject is given time to ask questions about the performance of the jumps. All tests are guided by the same examiner.

#### 3.2.2. Double-Leg Countermovement Jump (DL-CMJ)

The patient bents the knees until a semi-squat (knee ～90°), hips flexed, and then jumps explosively upward, trying to reach maximum height in the shortest possible time [37]. 

#### 3.2.3. Single-Leg Countermovement Jump (SL-CMJ)

The subject stands with one leg in the center of the plate and performs the jump as described above. The uninjured leg is tested first followed by the injured leg [37].

#### 3.2.4. Double-Leg Drop Jump (DL-DJ)

Starting from an upright position on the top of a vaulting box, the subject drops down from the box onto the center of the force plate considering the shortest possible ground contact time (<250 ms). This is followed by an immediately performed “vertical push-off action as quickly and as explosively as possible in order to perform the highest possible jump in the shortest possible time; thereby using an eccentric-concentric muscle action” [38]. The second landing is performed gently and with a stable leg axis on one leg in the center of the force plate.

#### 3.2.5. Single-Leg Landing (SL-Landing)

Starting from an upright position on the top of a vaulting box the patient takes a step forward and drops down onto the center of the 3D force plate. The landing is performed softly and with a stable leg axis on one leg. The patient must hold the position for five seconds without losing the balance before stepping off the force plate. The uninjured leg is tested first followed by the injured leg.

### 3.3. Jump Coordination

The jump coordination path is performed with the Speedy Basic Jump Set (TST Trendsport, Grosshöflein, Austria).

#### 3.3.1. Speedy Jump (SL-SJ)

The patient performs a coordination parkour as quick as possible without breaks. Then, 16 single-leg jumps through a course of red (forward-backward-forward jumps) and blue (sideways jumps) hurdles has to be completed. The hips have to be parallel during the performance. The contralateral leg is not allowed to touch the ground and the limbs are not allowed to touch the hurdle [36]. Time is taken with the stopwatch and the mean value is determined for each course. The sEMG records muscle activation, and 3D movement is tracked by the Azure Kinect DK.

#### 3.3.2. Quick-Feet-Test (SL-QFT)

The patient completes 15 steps in and out with one foot at a time, while being allowed to use the arms to keep the balance. A repetition is defined when the starting leg returns to the initial position. In case of a mistake or a step on the speedy pole, the test is stopped immediately [36]. Time is taken with a stopwatch and the tapping rate is counted. The sEMG records muscle activation, and 3D movement is tracked by the Azure Kinect DK.

### 3.4. Isometric Strength Tests

#### Gluteus Medius (G Med)

The patient lies supine with the pelvis fixed straight on an examination table, arms lay along the upper body, and the extremity to be tested is abducted by 30°. Starting with the uninvolved leg, the examiner fixes the expander two centimeters proximal to the lateral malleolus. No support from the hands or the opposite limb is allowed during the test. On command the patient pulls the leg to the side evenly and without momentum (4–6 s) while the tester fixes the expander with the help of a weight. It is important that only the muscle group to be tested, the abductor muscles of the hip, is activated. The sEMG records the muscle activation.

### 3.5. Questionnaire

The patients complete the International Knee Documentation Committee (IKDC), the ACL-Return to Sport Injury Scale (ACL-RSI) and the Tegner Activity Scale (TAS). The patient-reported results on symptoms, recovery and functionality of the knee are crucial and of great importance as they represent the patient’s subjective assessment in addition to the objective biomechanical analysis.

### 3.6. Biomechanical Measurements

#### 3.6.1. SEMG and MVC

Prior to electrode placement the skin is cleaned and degreased with alcohol. Surface electrodes with a two cm center-to-center distance are attached along the direction of the muscle fibers on the evaluated muscle bellies. The signals are registered with 16-bit accuracy at a sampling rate of 1500 Hz and stored for subsequent analysis using Noraxon DTS unit (Noraxon USA). The EMG signals are filtered with a Butterworth high-pass filter (cutoff frequency 10 Hz) and a low-pass filter (cutoff frequency 500 Hz). The sEMG signal from the evaluated muscles are measured continuously during the modified BIA test battery.

The collected data is normalized to a reference value. Maximum voluntary contraction (MVC) is performed separately for each muscle investigated before the test trials. The MVC contractions is performed against a static resistance. The parameters of each muscle are expressed as a percentage of its MVC (%MVC).

#### 3.6.2. 3D Force Plate

Kinematic data is recorded with the 3D force plate Bertec FP4060-08 (Bertec^®^, OH, USA). The piezoelectric force plate provides data on forces and moments in three dimensions (Fx, Fy, Fz, Mx, My, Mz). The Bertec assesses the forces and moments acting at the contact point where the force is applied.

#### 3.6.3. Motion Capture Camera Setup

We are using the Microsoft Azure Kinect DK, a 3D marker-less motion capture camera system, to track the participant’s maximum valgus and varus knee position during the landing maneuvers. For this study, three cameras capture motion data synchronously at 30 Hz. They are placed in a circle around the subjects with approximately 120° spacing (one frontal view and two back-side views), fixed on tripods 90 cm above the ground and placed approximately three meters from the circle center. The motion capture system is synchronized to the sEMG system and force plate using an illuminated ball controlled by the sEMG software. This ball is placed to be seen by one camera. From the color stream of this camera, we infer the ball’s on and off phases. Subsequently, the synchronization shift is calculated.

The Azure Kinect Body Tracking SDK delivers 3D joint positions and orientations for each joint segment. From this, 3D knee angles are obtained by regarding the left and right ankle, knee, and hip joint positions by calculating the angles between the hip-knee and knee-ankle vectors. Additionally, the local 3D joint segment orientations are used to calculate flexion and extension, abduction and adduction, and internal and external rotation. The resulting kinematic and angular data is filtered using a 4-th-order Butterworth filter with a cutoff frequency of 10 Hz.

Studies show good reliability when using the Azure Kinect for Motion analysis purposes [29]. So far, the Azure Kinect camera has been evaluated for gait analysis and slower functional movements [31]. Although the movement of subjects during the protocol is very fast, especially for the quick feet and countermovement jumps, we are only interested in the maximum varus and valgus knee angles during the landing phase. Therefore, a sampling frequency of 30 Hz is regarded as sufficient as we are not analyzing the entire complex functional movement. For this purpose, the camera would have to be evaluated against a marker-based motion capture system. Furthermore, compared to video camera systems, where only 2D information can be retrieved, the 3D system allows for the calculation of 3D angles and therefore provides more insightful data. Especially this marker-less motion capture offers huge advantages as the time-consuming marker placement can be alleviated and is thus easier to implement in everyday clinical practice.

## 4. Data Analysis and Planned Statistical Analysis

All data are presented in a descriptive manner, including anthropometric data and absolute measured values. Side-to-side differences are calculated for all SL tests, using the limb symmetry index (LSI). Therefore, absolute values of the ACL-reconstructed leg are divided by the values of the unaffected leg and multiplied by 100. For the stability, quick feet and speedy tests LSI is calculated by dividing the measured value of the unaffected leg by the value of the ACL-reconstructed leg and multiplied by 100 according to the calculation done in the study by Herbst et al. [17].

Power is quantified in the sagittal plane and defined through force and velocity. This parameter is directly extracted from the force plate data and reported as extracted. It is assessed for jumping during loading phase and total jump height. Power is based on peak power calculations for the lower extremity, which is defined in newton for maximal applied force during contraction. Additionally, it will be interpreted in the context of recent LSI-papers.

Descriptive quantitative data are calculated as the mean ± standard deviation. The Kolmogorov-Smirnov test is used to test the normality assumption. To compare the variances across the means of the three groups we use ANOVA repeated measures. Wilcoxon and Friedman-Test are used as well to calculate between-group differences.

All statistical analyses are performed using SPSS (IBM SPSS Statistics, New York, NY, USA). The significance level for all statistical tests is set a priori to 0.05. A post-hoc power analysis is conducted when significant effects are found.

## 5. Discussion and Expected Results

The purpose of the present study was to investigate the impact of orthotic devices and to analyze muscle activation and strength on functional stability of the knee during the modified test battery six months after ACLR.

It might be possible that soft braces or rigid orthoses improve functional performance during the modified BIA test battery. However, it is also possible that patients may be hindered in performing the test by wearing these aids, resulting in a decrease in functional performance.

As already mentioned, feedback and feedforward strategies play a major role in the completion of this test battery as well as in the general practice of sports [23,24].

We assumed that an ACL injury as well as ACLR negatively influence the mentioned feedback and feedforward strategies. Therefore, athletes may benefit from consistent and long-term use of soft braces or rigid orthoses at this time until they reach their initial performance level.

On the other hand, if the performance deteriorates due to the wearing of soft braces or rigid orthoses, the fundamental use of these aids in the rehabilitation process would have to be discussed and investigated more closely.

In addition to functional performance, the psychological aspect of wearing soft braces or rigid orthoses should also be considered.

Through to semitendinosus stripping with only partially regeneration of the tendon and postoperative immobility it might be possible that the sEMG results of the medial knee flexor show a strong imbalance of muscle activation patterns between the affected and unaffected limb and thus show worse results in the BIA test battery. In this case one could discuss the following aspect. Since the ischiocrural muscles act as agonist of the ACL function and as a medial stabilizer one could focus more on targeted neuromuscular training in order to strengthen these muscles. On the other hand, one could also discuss whether the use of other ACL grafts such as peroneus longus split could be an alternative to the semitendinosus graft as it has no primary influence on the biomechanics of the knee.

Ueno et al. found that increased knee adduction is associated with pelvic tilt and reduced gluteus medius strength and could therefore increase the risk of ACL injury [39]. Moreover, in a scoping review, Rinaldi et al. emphasized the impact of gluteal muscle strength on knee position during walking, running, jumping and landing [28]. However, Llurda-Almuzara et al. pointed out that gluteus medius and semitendinosus, among other muscles, showed no association between DKV and neuromuscular response [40]. Due to the controversial statements, this study aims to investigate on the influence of gluteus muscle strength and semitendinosus muscle activation and their possible influence on DKV and pelvic drop.

In summary, the obtained results will provide further information on the rehabilitation process. Accordingly, conclusions can be drawn, and training adapted to avoid reinjuries, e.g., due to muscular imbalances. However, if the tests provide excellent results, this would be a reason to establish jump tests earlier in the rehabilitation process in order to be able to release people earlier into their regular everyday life. For personal and economic reasons, this could be especially important for professional athletes so that they can hopefully return to their preinjury level as quickly as possible.

## Figures and Tables

**Table 1 healthcare-11-00513-t001:** Individual group test setup.

Run	Group A	Group B	Group C
1	orthosis	brace	no aid
2	brace	no aid	orthosis
3	no aid	orthosis	brace

**Table 2 healthcare-11-00513-t002:** Description of the planned consecutive activities during the measurement session [17,35].

**Participants**	
Signing informed consent and randomization ↓ IKDC, ACL-RSI, Tegner activity scale ↓ Placement EMG (both legs): Gluteus medius muscle and semitendinosus muscle ↓ Knee and hip range of motion measurement using standard goniometer ↓ Warm-up: 5 min. ergometer at 75 watts, 3 × 5 squats ↓ Putting on rigid orthosis or soft brace or leaving with no aid	
**Functional—Biomechanical Assessment**	
1. Stability tests Equipment: Zebris, sEMG Variables: level of stability, center of pressure, muscle activity	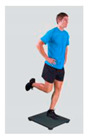 *Picture 1* Double-leg stability test → 1 rep for 10 s Single-leg stability test on the unaffected leg → 1 rep for 10 s Single-leg stability test on the ACL-reconstructed leg → 1 rep for 10 s
2. Jumps Equipment: 3D force plate Bertec, Azure Kinect, sEMG, vaulting box Variables: jump height (cm), power (W/kg), ground contact time (ms), reactivity (mm/ms), muscle activity, knee-angle	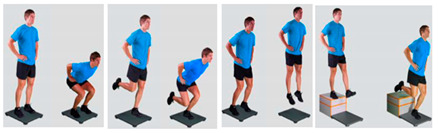 *Picture 2* Double-leg countermovement jump (CMJ) → 3 rep (4 s break in between) Single-leg CMJ on the unaffected leg → 3 rep (8 s break in between) Single-leg CMJ on the ACL-reconstructed leg → 3 rep (8 s break in between) Double-leg drop jump → 3 rep (8 s break in between) Single-leg landing on the unaffected leg → 3 rep (4 s break in between) Single-leg landing on the ACL-reconstructed leg → 3 rep (4 s break in between)
3. Jump coordination Equipment: speedy basic jump set, quick feet jump set, stopwatch, tapping rate counter, sEMG, Azure Kinect Variables: time (s), tapping rate, muscle activity, knee-angle	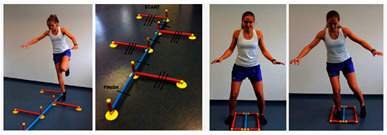 *Picture 3* Single-leg speedy jump on the unaffected leg → 1 rep Single-leg speedy jump on the ACL-reconstruced leg → 1 rep Double legged—quick feet test → 1 rep (15 circles, starting with the right leg)
4. Isometric hip abductor strength test (rt/lt) Equipment: sEMG, Expander Variables: strength (N) muscle activity	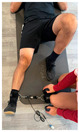 *Picture 4* unaffected leg → 1 rep for 5 s ACL-recontructed leg → 1 rep for 5 s

## Data Availability

Data is available from the authors on request from the corresponding author.

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
