# Peer review of "Protocol for a Randomized Crossover Trial to Evaluate the Effect of Soft Brace and Rigid Orthosis on Performance and Readiness to Return to Sport Six Months Post-ACL-Reconstruction"

_healthcare, 2023, doi:10.3390/healthcare11040513_

Round 1
Reviewer 1 Report
This manuscript describes a protocol for the biomechanical evaluation of different orthotic treatments after anterior cruciate ligament rupture. Overall, it is a very interesting manuscript with a comprehensible planned study design. I have no suggestions for corrections to the study design. However, there is no discussion of the evaluation methods presented. I would suggest shortening the introduction considerably and adding a discussion at the end of the manuscript. Here, many aspects of the introduction can be taken up and, in addition, the authors' own approach can be discussed with other protocols.
Author Response
Dear Reviewer,
please see the attachment.
Thank you very much.
Kind regards,
Sonja Jahnke

Reviewer 2 Report
Please see attached document.

Author Response

(The authors gave the same response as above.)

Reviewer 3 Report
Once the bolded part about motion capture in MM part of the protocol was changed, I believe this protocol is ready.
Author Response
Dear Reviewer,
Thank you for your comment. We have adjusted some details to further improve our protocol and hope that it is now suitable for publication.
Yours sincerely,
Sonja Jahnke
Round 2
Reviewer 1 Report
My suggestions for improvement have been fully implemented. I recommend the acceptance of the maunuscript.
Author Response
Dear Reviewer,
thank you for all your efforts! We appreciated your comments and are glad that we have implemented your suggestions to improve our protocol to your satisfaction.
Reviewer 2 Report
Please see attached file.

Author Response
Dear Sir or Madam,
please see the attachment.
Thank you very much.

Round 3
Reviewer 2 Report
Thank you for your time on these revisions and for your work on this paper.
Author Response

(The authors gave the same response as above.)
